# A Series of 40 Congenital Lung Malformation Cases and the Informative Value of CPAM Lesion Ratios

**DOI:** 10.3390/pediatric17010005

**Published:** 2025-01-09

**Authors:** Melanie Le, Phillip Harms, Kersten Peldschus, Carl-Martin Junge, Christian Tomuschat, Konrad Reinshagen

**Affiliations:** 1Clinic for Pediatric Surgery, University Medical Center Hamburg-Eppendorf, 20246 Hamburg, Germany; 2Department of Pediatric Radiology, University Medical Center Hamburg-Eppendorf, 20246 Hamburg, Germany; 3Department of Pediatric Radiology, Altona Children’s Hospital, 20246 Hamburg, Germany

**Keywords:** congenital lung malformation, congenital pulmonary airway malformations (CPAMs), lesion-to-lung ratio, pediatric surgery

## Abstract

Introduction: In this single-center retrospective analysis, we present case data and insights gathered over the past eight years. Additionally, we computed postnatal, pre-therapy lesion-to-lung ratios of Congenital Pulmonary Airway Malformations (CPAMs) to retrospectively assess potential outcome prediction using lesion-to-lung ratios. Methods: Data were collected between 2015 and 2022. Information such as chosen therapy, surgical duration, postoperative hospital stay, and follow-up was obtained from electronic case records. Pre-therapy pulmonary lesion volumes [mm^3^], lesion-to-ipsilateral-lung ratio, and lesion-to-both-lungs ratio of CPAMs were retrospectively calculated from computed tomography images using specialized software. Results: Of the 40 identified cases, 27 had CPAM, 7 had pulmonary sequestration, 4 had bronchogenic cysts, and 2 had congenital lobar emphysema. Histological examinations of resected specimens revealed no malignancy. For CPAMs, the median surgery age was 7 months (interquartile range (IQR): 0.45–11), averaging 9.54 ± 15.01 months. The CPAM surgery lasted on average 126 ± 53 min (median 124 min (IQR 108–172)). The mean length of hospital stay was 6 ± 1.41 days for thoracoscopic surgery and 17 ± 18.23 days for open surgery. No clear link was found between the lesion ratio and management choice or surgical length. Notably, larger lesions tended to result in longer postoperative stays. CPAMs with a lesion-to-ipsilateral-lung ratio of ≤0.11 were asymptomatic. Discussion and Conclusions: If patients present no symptoms, mild symptoms, or smaller CPAM lesions, “wait and watch” and a CT scan of the thorax up to approximately six months of age remain a reasonable approach. The true risk of malignancy remains ambiguous, especially since there was no evidence of malignancy in our biopsies. On the other hand, prophylactic surgery before symptoms arose led to earlier discharge and overall low intraoperative complications compared to symptomatic counterparts. Ultimately, the adopted therapy pathway remains a parental choice. For CPAMs, an increased lesion-to-lung ratio correlated with extended hospital stay and symptomatic presentation. However, there was no cut-off value for conservative or surgical treatment.

## 1. Introduction

Congenital lung malformations (CLMs) are a heterogenous group of rare abnormalities involving the parenchyma, the tracheobronchial system, and the pulmonary vessels. The most frequently observed malformations include Congenital Pulmonary Airway Malformations (CPAMs), forming approximately 30% of all CLMs, intra- and extralobar pulmonary sequestration (BPS), congenital lobar emphysema (CLE), and bronchogenic cysts (BCs). A notable characteristic of CLMs is the heightened risk of recurrent bronchopulmonary infections and, for some subtypes, the potential of malignant transformation. Furthermore, CPAMs and CLE may lead to early pulmonary decompensation requiring emergency interventions. While surgery has been advocated for decades for symptomatic CLMs, evidence for the treatment of asymptomatic patients is scarce and is currently investigated in a prospective, multicentric study [1]. Surgeons advocating for prophylactic early surgery in all CLMs refer to the potential of compensatory postoperative lung growth and the prevention of malignancies, which may occur at any age. However, the precise incidence of malignancy for CLM types remains ambiguous [2]. Improved imaging techniques have led to an increasing number of CLM diagnoses, especially regarding antenatal detection. Furthermore, some centers proceed with fetal MRI after antenatal ultrasonographic diagnosis of CPAMs, to further specify the volume ratio as an indicator to guide surveillance frequency and mediastinal shift. However, most centers follow the algorithm of postnatal cross-sectional imaging of CLMs.

Regarding current controversies, we aimed to share our experiences with CLM management, predominantly concerning CPAMs, and the effect of postnatal lesion ratios on therapeutic decision-making and the clinical course.

## 2. Materials and Methods

A retrospective, single-center clinical chart review was conducted on patients treated for congenital lung abnormalities at the University Medical Center Hamburg-Eppendorf (UKE) and Altona Children’s Hospital (AKK). Selected ICD-codes were used to identify cases (Q33.0; Q33.1; Q33.2; Q33.3; QQ33.4; Q33.5; 33.6; Q33.8; Q33.9; C34.0; C34.1; C34.2; C34.3; C78.0; C78.2; D14.3; J90). All CLM patients (0 months–18 years) treated at both institutions between 2015 and 2022 were included in the analysis. Demographic information was obtained from electronic case records, including age at diagnosis, clinical presentation, age at surgery, surgical technique, time from incision to suture, intraoperative and postoperative complications, length of hospital stay, and subsequent clinical course. The pre-therapy pulmonary lesion volumes [mm^3^], lesion-to-ipsilateral-lung ratio, and lesion-to-both-lungs ratio of CPAMs were calculated by the pediatric radiology department of our center using the specialized software Philips IntelliSpace Portal 12.1, an image post-processing platform that semi-automatically calculates the diameter and volume of lung lesions using the children’s computed tomography (CT) images. CT images were extracted from in-house electronic case records. The gathered data were presented and analyzed descriptively.

## 3. Results

A total of 40 cases were identified; of these, 27 were CPAMs, 7 were BPS, 4 were bronchogenic cysts, and 2 were CLE (Figure 1A). A total of 25 CLMs were treated surgically. A histological examination was performed for each resection (n = 25), showing no signs of malignancy.

### 3.1. CPAM Cohort

Of the CPAM cases, 70.4% (19/27) underwent surgery and 25.9% (7/27) were conservatively managed (watch and wait) (Figure 2A). Of the 27 children, 16 were male and 11 were female. Four patients had one or more associated malformations, with cardiac malformations (n = 5) being the most common (ventricular septal defect, dextroversion, persistent foramen ovale, patent ductus arteriosus, and coarctation of the aorta) and two renal malformations (nephroblastoma and pyelectasis). The most common CPAM type (Stocker classification; cyst diameter < 2 cm) was type 2, accounting for 72,2% of all CPAM cases (Figure 1B). Regarding the postnatal symptoms, in our cohort CPAMs were most frequently associated with mediastinal shift (35.7%), impaired postnatal adaptation (28.6%), defined as tachypnea, thoracic retractions, and oxygen or continuous positive airway pressure (CPAP). Recurrent infections were present in 21.4% and pneumothorax in 14.3% of the cases (Figure 2B). Out of 19 (57.9%) operated CPAMs, 11 had preoperative symptoms. The average age at surgery was 9.54 ± 15,01 months; median = 7 months (interquartile range (IQR) 0.45–11). The mean age for symptomatic patients at surgery (n = 11) was 4.42 ± 6.53 months, median 0,84 months (IQR 0.17–9) versus 15.31 ± 21.79 months, and median 8 months (IQR 7–13) for asymptomatic patients (n = 8) (Figure 2B). No relevant intraoperative or postoperative complications occurred with either surgical technique (open and thoracoscopic), with the most common complication being a pleural effusion and pneumothorax.

The mean duration of surgery was 126 ± 53 min (median 124 min (IQR 108–172)) in the CPAM cohort (Figure 3). Only two cases were treated entirely with thoracoscopy. The thoracoscopic procedure resulted in a mean operative time of 88 ± 55 min versus 138 ± 42 min (median 125 min (IQR 110–174.5)) for the open approach. The average length of hospital stay was 6 ± 1.41 days vs. 17 ± 18.23 days for thoracoscopic and open approaches. In the surgical cohort, preoperative asymptomatic patients with prophylactic surgery were discharged earlier compared to their symptomatic peers (mean 6 ± 1.50 days (median 6 days (IQR 5–7.5)) vs. 23 ± 20.85 days (median 17 days (IQR 8–29.5)), as shown in Figure 2B. Overall survival of 100% at 1 and 5 years was achieved in all patients with CPAMs, regardless of the treatment regimen or the chosen surgical procedure. One patient who was treated surgically continued to show recurrent infections postoperatively (>2 infections/year). The follow-up for the surgical cohort ranged from 3 months to 24 months with a mean of 7.46 ± 5.44 months (median 8 months (IQR 2–10.5)). The conservatively treated patients had a mean follow-up of 10.49 ± 5.23 months (median 8 months (IQR 8–12.5)). Two patients with conservative treatment presented recurrent infections.

### 3.2. CPAM CT Findings

Chest CT imaging was available for 17 CPAM patients. Two cases were excluded from the analysis due to pneumothorax and lung agenesia limiting the evaluation of the CPAMs. Hence, 15 patients were included for the analysis of the CPAM volumes (Table 1). In two cases (cases No. 1 and 2, Table 1), the lesions were too large in scale and could not be segmented by the software but were still included in the analysis for comprehensiveness (one case presented in Figure 4).

The mean measurable lesion volume [mm^3^] was 39.54 ± 42 mm^3^ with the smallest lesion measuring 2.4 mm^3^ and largest lesion being 145 mm^3^. To determine the relationship between lesion size, age, and body size, we calculated two ratios: lesion-to-ipsilateral-lung ratio (lesion size [mm^3^] divided by the ipsilateral lung size [mm^3^]) and lesion-to-both-lungs ratio (lesion size [mm^3^] divided by the total lung size including the left and right lungs [mm^3^]). The average ratio was 0.28 ± 0.44 with a range from 0.03 to 1.74. The case with the highest lesion-to-ipsilateral-lung ratio (No. 3, Table 1) had the longest length of stay at 17 days. The longest hospital stay was 70 days in cases that could not be measured using software segmentation because it was too large in scale. In patients who underwent conservative management, the average lesion-to-ipsilateral-lung ratio in operated cases was 0.34 ± 0.57 and 0.18 ± 0.12 in patients with conservative management. Two patients with unmeasurable lesions were included in this cohort. Furthermore, these two cases presented the longest length of stay of 70 and 42 days, respectively. The patient (No. 4, Table 1) with the highest measurable lesion volume (145 mm^3^) and a rather high lesion-to-ipsilateral-lung ratio of 0.34 was conservatively treated (Figure 5). However, in this case, the parents decided against surgical treatment despite a clear recommendation for surgery. The patient continued to show recurrent infections. Retrospectively, there was no clear pattern for therapy management (surgery or watchful waiting) regarding lesion volume or for both ratios. Furthermore, there was no clear trend when correlating surgery time with lesion volume or both ratios.

## 4. Discussion

The optimal treatment strategy for asymptomatic pediatric patients with small congenital pulmonary abnormalities remains much discussed. Some surgeons advocate the resection of all lesions, regardless of size, to minimize the risk of subsequent malignancy development [1]. Numerous studies have shown that compensatory growth in the unaffected lung is observed when surgery is performed early in life [3]. In the context of CPAMs, literature reviews suggest that surgical interventions performed between 3 and 6 months of age are technically less demanding than those conducted later in life because of the absence of prior chest infections [4,5]. Surgeries for children older than 9 months have been associated with significantly longer operative durations, potentially indicating an increased technical complexity [3]. However, some clinicians advocate a watchful waiting approach, claiming that the true incidence of adverse outcomes is not well established [2]. The risk of malignancy might not be sufficiently high to justify subjecting an asymptomatic child to a major operation; therefore, a nonoperative strategy could be more appropriate until adverse signs emerge [4]. In the current study, all biopsies of CLMs showed no signs of histopathological malignancy. However, it is possible that malignant transformation had not yet occurred during this period and thus was not histopathologically detectable due to the relatively young age at the time of surgery (mean ages at surgery: 9.54 ± 15.01 months with a median of 7 months (IQR 0.45–11) for CPAMs; 10.47 ± 5.79 months with a median of 12 months (IQR 7–15) for BPS). A systematic review of tumor-associated congenital pulmonary malformations, encompassing 134 publications, revealed diagnoses of malignancy at later age stages, with a mean age of surgery at 3.68 ± 3.4 years [2].

All in all, the presence of symptoms alone does not ultimately imply surgical treatment. The overall setting of the patient must be evaluated, such as the severity of symptoms with improvement or worsening in the follow-up. The final decision for surgery was ultimately made by the parents after extensive counseling. Small asymptomatic lesions were surgically treated, often as requests of the parents due to the risk of malignancy. Retrospectively, a conservative approach for small asymptomatic lesions could have been chosen, since none of the biopsies showed evidence of malignancy.

### 4.1. Surgical Management

Surgical intervention demonstrated an overall low incidence of complications both during and after the procedure. Two surgeons were primarily involved in all the CPAM operations, which may have influenced the operation time. Our investigation identified mild self-resolving pneumothorax as the most common postoperative consequence in patients with CPAMs. Asymptomatic CPAM patients had a higher mean age at surgery (15.31 ± 21.79 months) than their symptomatic counterparts (4.42 ± 6.53 months) (Figure 2B), as the need for intervention was established earlier in symptomatic patients. In asymptomatic patients, the decision to operate is often made electively in a follow-up examination. Comparing the symptomatic and the asymptomatic cohort after surgical intervention, there is a trend for shorter postoperative hospital stays and overall lower intraoperative complications for patients without preoperative symptoms (Figure 2B). Hence, prophylactic surgery before symptoms arise seems to be beneficial in terms of earlier discharge and intraoperative complications. Regarding the open or thoracoscopic approach, our results demonstrate that the thoracoscopic approach leads to shorter operative times and that earlier discharge shortened hospital stays (88 ± 55 min vs. 138 ± 42, 6 ± 1.41 days vs. 17 ± 18.23 days for CPAMs; 89 ± 33 min vs. 195 ± 109 min; 3 ± 2.51 days vs. 15 ± 6.36 days for sequestration). Although the benefits of thoracoscopic surgery are evident, it is not always a viable option and must be carefully assessed in terms of patient safety. In two cases, an accurate distinction between healthy and pathological tissue was unattainable, necessitating a subsequent open thoracotomy following the initial thoracoscopy. It is advised that smaller, well-defined lesions are addressed with thoracoscopic surgery, whereas larger or sub-optimally positioned anomalies are best addressed with open surgery. However, considering the advantages of thoracoscopy, it can be assumed that future data will show an increase in thoracoscopic interventions. Although CPAMs represent primary congenital pulmonary malformations at our institution, an average of 2.37 surgical interventions for CPAMs are performed annually. The annual frequency of CPAM operations remained constant throughout the recorded period, with no observable increase in surgical interventions attributable to advancements in prenatal ultrasonography diagnostic capabilities.

### 4.2. CPAM CT Findings

By calculating lesion-to-ipsilateral-lung ratios and lesion-to-both-lungs ratios, we related lesion size to total lung size to account for age and body size. As illustrated in Table 1, more pronounced clinical manifestations are associated with larger CPAM lesions, whereas smaller lesions, characterized by lower lesion-to-lung ratios, predominantly exhibited no symptoms. Lesions with a lesion-to-ipsilateral-lung ratio of <0.11 did not exhibit symptoms. Our findings did not reveal an association between lesion volume and the selected management approach (surgical intervention or observation) or length of surgery. There was no cut-off value for surgery or “wait and watch”. In the surgery cohort, we found a trend toward an extended length of postoperative hospital stay for increased lesion volume. Early hospital discharge seems to be facilitated by smaller lesion ratios and asymptomatic clinical presentation. Furthermore, some authors have questioned whether postnatal CT is necessary when prenatal MRI is available [6]. Unfortunately, since most of our patients were referred from outside clinics, prenatal ultrasound findings and prenatal MRIs were unavailable. Approximately 25% of initially asymptomatic patients with antenatally diagnosed CPAMs ultimately develop symptoms, typically presenting around the age of 6–7 months [1]. As a result, it is advised that clinical re-evaluation occurs at the 6-month mark [7,8]. At least one postnatal CT scan should be performed to determine lesion type and facilitate preoperative planning.

### 4.3. Limitations

Our study offers valuable insights into congenital pulmonary malformations by focusing on the 40 cases examined. Notably, much of our data pertain to CPAMs, as the counts for other malformation subtypes were limited, thus challenging the broad conclusions. Some patients chose to continue their follow-up at their original clinics. Although it would be interesting to assess lung function as a follow-up, the age of our young patients makes such evaluations challenging. Our study included 19 surgical interventions in 27 CPAM cases, representing only a snapshot of this medical landscape. Further investigations using more extensive datasets are necessary for a more comprehensive and predictive understanding.

## 5. Conclusions

The presence of symptoms alone does not ultimately imply surgical treatment. Surgical intervention should always be meticulously tailored to the patient’s unique circumstances, factoring in the specifics of their medical situation and, importantly, parental preferences. If patients present no symptoms, mild symptoms, or smaller CPAM lesions, “wait and watch” and a CT scan of the thorax up to approximately six months of age remain a reasonable approach. The conservative approach is supported by the fact that none of our biopsies showed signs of malignancy. Regarding lesion ratios, we observed that higher lesion-to-lung ratios tend to correlate with extended postoperative hospital stays. However, there was no cut-off value for surgical or conservative management.

## Figures and Tables

**Figure 1 pediatrrep-17-00005-f001:**
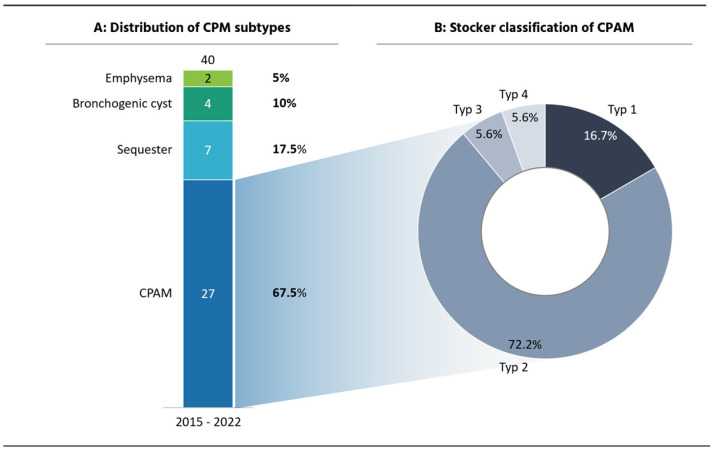
Overview of congenital pulmonary malformations. (**A**) Distribution of CPM subtypes. Forty congenital pulmonary malformations were identified between 2015 and 2022 (eight years). CPAMs were the most common malformation (n = 27). (**B**) Stocker classification for CPAMs based on cyst size and histomorphology. Type 1: medium-to-large interconnecting cysts (1–10 cm). Type 2: bronchiolar-like cysts (0.5 to 2 cm) that blend with normal parenchyma. Type 3: solid mass involving the lobe or even the entire lung. Type 4: cysts are distributed peripherally, can be multiple, and involve more than one lobe. The most common type of CPAM used in this study was type 2 (72.2%).

**Figure 2 pediatrrep-17-00005-f002:**
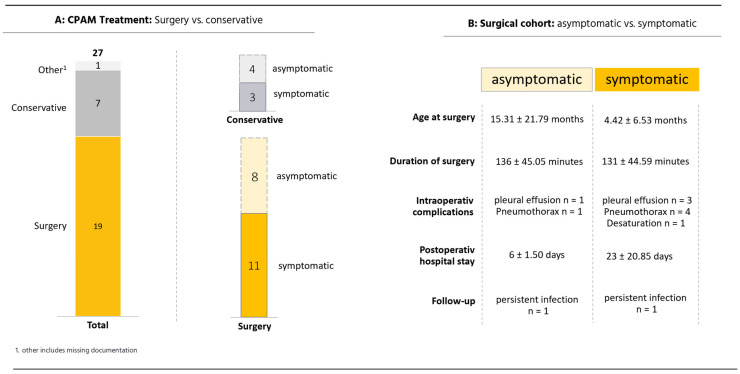
CPAM treatment; symptom distribution. (**A**) CPAM treatment: Surgery versus conservative treatment. Left bar chart: Of all CPAM cases (n = 27), 19 were treated with surgery (yellow) and 7 by observation (grey). One case lacked further documentation after the initial consultation (light grey). Right bar chart: Of the 27 CPAM cases, 14 patients exhibited symptoms. Three symptomatic cases were treated conservatively (gray), and 11 symptomatic cases were operated upon (yellow). (**B**) Surgical cohort: Asymptomatic vs. symptomatic. Asymptomatic and symptomatic patients who were treated by surgery compared for age at surgery [months], duration of surgery [minutes], intraoperative complications, postoperative hospital stay [days], and complication presented in the follow-up.

**Figure 3 pediatrrep-17-00005-f003:**
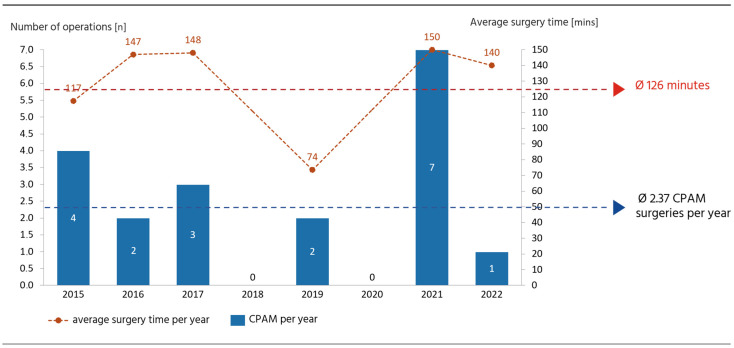
Number of operations and average surgery time for CPAMs. Left *y*-axis: number of operations [n]. Right *y*-axis: average surgery time in min [min]. *X*-axis: 2015–2022. Blue bars represent the number of CPAM operations per year. Blue dotted arrow: average number of CPAM surgeries per year. Red dots represent the average surgery time for each year. Red dotted arrow: average surgical time including all CPAM surgeries.

**Figure 4 pediatrrep-17-00005-f004:**
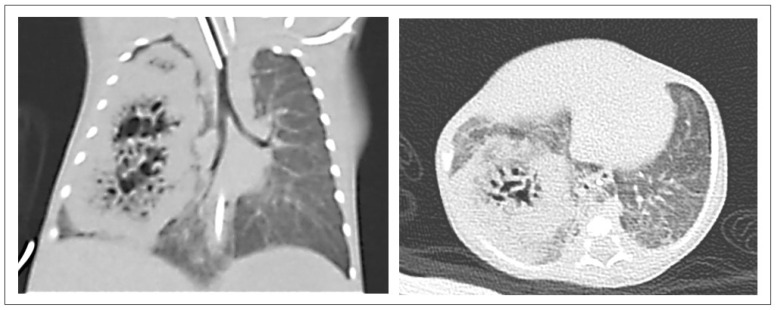
CT-Thorax (coronar, transversal). CPAM lesion not measurable by segmentation (too large in scale). CT-Thorax of Case No. 2 in Table 1 at the age of 20 days. The patient required intubation for postnatal respiratory insufficiency. This CPAM, type 2, involved nearly the complete right hemithorax. The patient underwent a right-sided thoracotomy with resection of the upper and middle lobes. Lesion volume: not measurable.

**Figure 5 pediatrrep-17-00005-f005:**
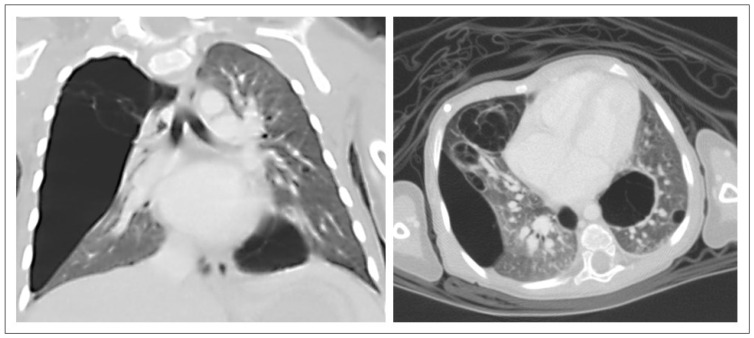
CT-Thorax (coronar, transversal). Large bilateral CPAM lesions, conservatively treated. CT of the Thorax of Case No. 4 in Table 1 at the age of 10 months. Bilateral CPAM type 4 in all four lung quadrants. The patient presented postnatal pneumothorax that required intubation and thoracic drainage. Despite a clear recommendation for surgery, the patient’s parents decided against surgical treatment. Lesion volume: 145 mm^3^; lesion-to-ipsilateral-lung ratio: 0.34; lesion-to-both-lungs ratio: 0.34.

**Table 1 pediatrrep-17-00005-t001:** CPAM CT-Volumes: lesion volume [mm^3^]; lesion-to-ipsilateral-lung ratio; lesion-to-both-lungs ratio.

Case No.	Lesion-to-Ipsilateral-Lung Ratio	Lesion-to-Both- Lungs Ratio	Lesion [mm^3^]	Age at CT	Symptoms	Therapy	Surgery Time [mins]	Length of Stay [days]
**1**	**Not** **measurable ***	Not measurable *	Lesion too large *	1 month	Postnatal respiratory adaptation disorder, mediastinal shift	Surgery	98	42
**2**	**Not** **measurable ***	Not measurable *	Lesion too large *	20 days	Respiratory insufficiency with need for intubation	Surgery	129	70
**3**	**1** **.74**	0.64	68	5 days	Mediastinal shift, tachypnoea	Surgery	108	17
**4**	**0** **.34**	0.34	145	10 months	Perinatal pneumothorax, repeated pulmonary infections	Observance	-	-
**5**	**0** **.29**	0.20	56	2 months	Postnatal respiratory adaptation disorder with sufficient development in further course	Observance	-	-
**6**	**0** **.26**	0.20	18.9	6 months	-	Surgery	89	5
**7**	**0** **.18**	0.11	39	7 months	-	Surgery	119	6
**8**	**0.18**	0.11	22.1	8 months	-	Surgery	177	8
**9**	**0.14**	0.08	101.9	6 years	One singular respiratory infection with need for antibiotics	Observance	-	-
**10**	**0.12**	0.06	15.8	13 months	Recurrent respiratory infections	Surgery (thoracoscopy)	49	7
**11**	**0.12**	0.07	10	6 months	-	Observance	-	-
**12**	**0.11**	0.05	9.4	7 months	-	Surgery	172	9
**13**	**0.09**	0.05	20	13 months	-	Surgery	210	6
**14**	**0.07**	0.04	5.5	8 months	-	Surgery (thoracoscopy with open conversion)	85	7
**15**	**0.03**	0.01	2.4	6 months	-	Observance	-	-

Fifteen CPAM patients with computed tomography (CT) images were included in this study. In each case, the absolute lesion size [mm^3^] was calculated using the Philips IntelliSpace Portal 12.1. Furthermore, two ratios were calculated: for lesion-to-ipsilateral-lung ratio, the lesion volume [mm^3^] was divided by the volume of the total ipsilateral lung [mm^3^]. For lesion-to-both-lungs ratio, the lesion volume [mm^3^] was divided by the total volume of both lungs [mm^3^]. The cases were arranged by numbers (“No.” as short for “number”), 1–15, in vertical order according to their lesion-to-ipsilateral-lung ratio. * The lesions in case No. 1 and 2 were not measurable (lesions were too large in scale).

## Data Availability

All relevant data are within this paper.

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
