# Peer review of "A Series of 40 Congenital Lung Malformation Cases and the Informative Value of CPAM Lesion Ratios"

_pediatrrep, 2025, doi:10.3390/pediatric17010005_

Round 1
Reviewer 1 Report
Comments and Suggestions for Authors
Because the age numerical data (e.g. age, duration of surgery, length of hospital stay) show great variation, it is advisable to present them as median and interquartile interval.
The following phrase should be clarified (rows 88-92):
"Four patients presented with one or more associated malformations (cardiac malformation, n = 5 (ventricular septal defect, dextroversion, patent foramen ovale, patent ductus arteriosus, and coarctation of the aorta]) and renal malformation (n = 2 (nephro-90 blastoma, pyelectasis])."
The following phrase should be clarified (rows 260-262):
"Asymptomatic CPAM patients had a higher a mean age at surgery (15,31 ± 21,79 months) compared to their symptomatic counterparts (42 ± 6,53 months) (Figure 2B) since symptomatic patients receive treatment more promptly."
Author Response
Dear Reviewer,
we thank you very much for your comments. We really appreciate your time and effort reading and reviewing our manuscript.
Comment 1: Because the age numerical data (e.g. age, duration of surgery, length of hospital stay) show great variation, it is advisable to present them as median and interquartile interval.
Response 2: Thank you very much for your advice. We have added the median and interquartile ranges into our results.
Comment 2: The following phrase should be clarified (rows 88-92): "Four patients presented with one or more associated malformations (cardiac malformation, n = 5 (ventricular septal defect, dextroversion, patent foramen ovale, patent ductus arteriosus, and coarctation of the aorta]) and renal malformation (n = 2 (nephro-90 blastoma, pyelectasis])."
Reponse 2: Thank you for your comment. That sentence was worded unclear. We corrected it accordingly: “Four patients had one or more associated malformations, with cardiac malformations (n=5) being the most common (ventricular septal defect, dextroversion, persistent foramen ovale, patent ductus arteriosus and coarctation of the aorta) and two renal malformations (nephroblastoma, pyelectasis).”
Comment 3: The following phrase should be clarified (rows 260-262): "Asymptomatic CPAM patients had a higher a mean age at surgery (15,31 ± 21,79 months) compared to their symptomatic counterparts (42 ± 6,53 months) (Figure 2B) since symptomatic patients receive treatment more promptly."
Response 3: Thank you very much for your comment. We corrected that sentence accordingly: “Asymptomatic CPAM patients had a higher mean age at surgery (15.31 ± 21.79 months) than their symptomatic counterparts (4.42 ± 6.53 months) (Figure 2B), as the need for intervention was established earlier in symptomatic patients.”
Reviewer 2 Report
Comments and Suggestions for Authors
Dear Authors,
thank you very much for your well-written manuscript, providing a descriptive analysis of the clinical presentation, treatment and follow-up in a cohort of lung malformation cases, which was treated in your pediatric departments in the past eight years. Please pay attention to the following comments and questions, pertaining to your manuscript:
1. Line 68. Your cohort was selected between the years 2015-2022. Please explain why not earlier, in order to provide a longer follow-up (>5 years) after their treatment.
2. Was there any technical evolution in the diagnosis and treatment in this field that may have influenced the decision making and therefore the clinical results of your cases?
3. Line 74. You analyzed the cases using a specialized software Philips IntelliSpace Portal. Please provide some additional information about this software: what kind of imaging analysis is performed, why necessary additionally to the computed tomography (CT).
4. Line 80. A total of 25 CLMs were identified in this study. You probably mean that a total of 40 cases were identified and 25 were surgically treated. Please correct.
5. Line 118. Chest CT imaging was available for 17 of the overall 27 CPAM patients. How did you diagnose and treated 10 cases without a postnatal CT imaging?
6. Line 190. 2 CPAM cases were excluded from your analysis due to pneumothorax and lung agenesia and 2 cases had unmeasurable lung volumes. Why not excluding all 4 cases from your analysis due to their insufficient radiological assessment?
Best Regards
Author Response
Dear Reviewer,
we thank you very much for your comments. We really appreciate your time and effort reading and reviewing our manuscript.
Comment 1: Line 68. Your cohort was selected between the years 2015-2022. Please explain why not earlier, in order to provide a longer follow-up (>5 years) after their treatment.
Response 1: Thank you very much for your comment. The electronic patient file and surgical documentation system was fully introduced in our center 2015. In order to ensure complete and uniform data collection, we have not included patients with paper records.
Comment 2: Was there any technical evolution in the diagnosis and treatment in this field that may have influenced the decision making and therefore the clinical results of your cases?
Response 2: Thank you for your comment. The overall trend is towards minimally invasive surgery, especially considering the advantages of thoracoscopy. We would assume that more recent and future data would show an increase in thoracoscopic interventions, compared to the fact that our study had very few thoracoscopic procedures. We added that interesting aspect into our manuscript.
Comment 3: Line 74. You analyzed the cases using a specialized software Philips IntelliSpace Portal. Please provide some additional information about this software: what kind of imaging analysis is performed, why necessary additionally to the computed tomography (CT).
Response 3: Thank you for your comment. That software is an image post-processing application, in this study used to objectively quantify lesion diameter and volume using the CT images. We have added this to the manuscript.
Comment 4: Line 80. A total of 25 CLMs were identified in this study. You probably mean that a total of 40 cases were identified and 25 were surgically treated. Please correct.
Response 4: Thank you for your comment. We corrected that typing error accordingly.
Comment 5: Line 118. Chest CT imaging was available for 17 of the overall 27 CPAM patients. How did you diagnose and treated 10 cases without a postnatal CT imaging?
Response 5: Thank you for your comment. Some patients did bring CT-images from external institutions, which could not be processed by our software. In the other cases, prenatal MRIs in combination with sonography and X-Rays were deemed sufficient for diagnosis and treatment decision making.
Comment 6: Line 190. 2 CPAM cases were excluded from your analysis due to pneumothorax and lung agenesia and 2 cases had unmeasurable lung volumes. Why not excluding all 4 cases from your analysis due to their insufficient radiological assessment?
Response 6: Thank you very much for your comment. We considered “too large for measurement” also to be a kind of definition for lesion size, with which the other parameters such as symptoms and duration of surgery can be correlated. We deemed cases with more parts of “lung agenesia” and “pneumothorax” compared to CPAM volume were comparable.
Reviewer 3 Report
Comments and Suggestions for Authors
The treatment of congenital pulmonary airway malformations remains unresolved due to their rarity, diversity, and the young age of affected patients, making large prospective clinical trials unlikely.
As a result, any new information from a relatively large patient group is valuable, even if this study does not provide a clear solution.
I have no specific comments on the study or manuscript; the objectives are clear, the methods and results are well presented, and the discussion aligns with the findings, supported by relevant clinical examples.
Author Response
Dear Reviewer,
we thank you very much for your comments. We really appreciate your time and effort reading and reviewing our manuscript.
Round 2
Reviewer 2 Report
Comments and Suggestions for Authors
Dear Authors,
thank you for providing comprehensive and convincing answers to the questions and queries expressed by me and the other Reviewers and made changes, that have contributed to the optimization of your manuscript and increased the publishing potential of your work. I have no further questions and queries, pertaining to your manuscript.
Best Regards